# Relative age of youth swimmers and their sporting performance at the end of the season

Mendoza-Castejón D[1,2ʘ‡], Trinidad A [2,3ʘ‡*], De la Calle L.M[1,2ʘ‡], Belando-Pedreño N [4ʘ‡]

1 Department of Sports Sciences, Faculty of Medicine, Health and Sports, Universidad Europea de Madrid, Villaviciosa de Odón, Madrid, Spain, 2 Aqualab Research Group, Universidad Europea de Madrid, Villaviciosa de Odón, Madrid, Spain, 3 Department of Education and Educational Innovation, Faculty of Law, Education and Humanities, Universidad Europea de Madrid, Villaviciosa de Odón, Madrid, Spain, 4 Department of Educational Sciences, Faculty of Teacher Training and Education, University of Oviedo, Oviedo, Spain

ʘ These authors contributed equally to this work.
‡ MCD, TA, DCLM and BPN also contributed equally to this work.
* alfonso.trinidad@universidadeuropea.es

## Abstract

This study explores the influence of relative age on the athletic and academic performance of young swimmers, while also considering other contributing factors such as training conditions, anthropometric characteristics, and coaches' subjective evaluations. A descriptive, explanatory, and prospective design was employed, using quantitative (questionnaires) and observational methods. The sample consisted of 33 national-level swimmers (11 males and 22 females). Variables analyzed included sex, date of birth, training data, academic performance (AP), final sport performance (SP), coaches' perceptions of daily performance, and anthropometric measurements. Results indicated no direct effect of relative age on the main variables. However, ANCOVA revealed significant differences based on birth quartile (p = .001), month of birth (p = .001), training frequency (p = .003), and body weight when mediated by sport category. Additionally, significant associations were found between relative age and sport performance when BMI was included as a covariate (p = .036), along with year and month of birth (p = .038; p = .027). Coaches' perceptions of performance were also significantly related to competitive category (p = .033). It is concluded that while relative age may influence athletic performance, its effect appears to be mediated by contextual and individual factors related to the athlete's preparation and environment.

## Introduction

The concept of relative age has been described in the sports domain as the cutoff date among individuals born in the same calendar year, aiming to achieve competitive equity [1]. This calculation has allowed athletes to be grouped into three (Q1-Q3) [2] or four quartiles (Q1-Q4) [3]. For the latter, the distribution is as follows: the

**Data availability statement:** All relevant data are within the paper and its Supporting Information files.

**Funding:** The author(s) received no specific funding for this work.

**Competing interests:** The authors have declared that no competing interests exist.

first quartile includes those born between January and March; the second quartile, between April and June; the third quartile, between July and September; and the fourth quartile, between October and December [4]. However, this intentional grouping practice can often favor relatively older athletes and disadvantage relatively younger ones within the same age group [5,6]. The consequences arising from these differences are known as "relative age effect" (RAE) [7].

Among these effects, the scientific literature corroborates the existence of significant differences concerning the athlete's context or cultural environment [8], as well as the competency base, the degree of early specialization, and the diversity of the sport modality [1]. Additionally, other differentiating aspects are noted, such as individual characteristics, gender or age group, competitive category or level, and type of sport or event) [9]. Furthermore, the significant effects of relative age have also been extrapolated to other areas such as education, where differences among students have been found due to interactions between gender, age, and academic performance in Physical Education (PE), with enjoyment of physical activity ($r = 0.28$), motor self-efficacy ($r = 0.27$), and the level of physical activity (PA) performed by students ($r = 0.21$) [10].

Regarding sports practice, various studies have also corroborated that one of the more significant effects of relative age is found in strength and endurance sports on their performance, but not in more technical sports [11,12] or individual disciplines like swimming. High technical demands of swimming have also led to studies on the effects of relative age on competitive performance. It has been evidenced that the swimmer's body size and the amount of propulsive force they can generate during the swimming test through strokes are conditioning factors [13]. Additionally, the scientific literature identifies other significant effects that could impact performance, such as the technical ability to reduce drag [14], the development of maximum and explosive strength [15,16], maximum oxygen consumption [17], aerobic endurance [18], and aerobic metabolism [19], along with anthropometric differences among swimmers [20].

But similar to other sports, swimming also involves the detection of potential talents and the undervaluation of athletic potential because of relative age [5,6]. Younger swimmers and those with late development would be at a competitive disadvantage compared to their predecessors [9]. These circumstances would sometimes lead to sports dropout [21] until these swimmers reach growth deceleration and [22,23]. Consequently, they would have greater progression in anthropometric and physical attributes compared to early-maturing athletes during adolescence and adulthood [24,25].

Considering these arguments and as a contributing factor to athletic performance in young ages where the athlete's development is being established, the coach's interpersonal style and the training climate during sessions are crucial [26]. The way daily training sessions are implemented and guided has the potential to influence physical, technical, tactical, and psychological development, as well as long-term success [27]. A highly trained, qualified, and experienced coach would be a guarantee of success and would enhance athletes' performance [28]. Therefore, coaches

play a crucial role in identifying the specific potentials and needs of their athletes. These factors suggest that, in individual and technically complex sports such as swimming, there is a need to explore how RAE interacts with technical performance variables such as water efficiency, motor coordination, and biomechanical adaptation, in addition to anthropometric and physiological differences. Additionally, studies exploring the role of the training environment — particularly the subjective perceptions of coaches and the conditions in which they practise — as mediating factors in the impact of RAE on the development of young swimmers are lacking.

Finally, considering the reviewed literature, the main objective of the present study was to analyze the relationship between RAE and sport performance achieved at the end of the season in competitive swimmers. Additionally, it aimed to examine the relationship of RAE with the anthropometric profile, training conditions, and coaches' subjective perception of swimmers' daily performance. Furthermore, it sought to review how these variables could affect sporting and academic performance of these young swimmers. We hypothesise that RAE affects the athletic and academic performance of swimmers, their physical profile, their training conditions, and how their day-to-day performance is perceived by coaches.

## Materials and methods

### Participants

Thirty-three national-level swimmers 11 males and 22 females (15±2.08 years and 51.5±6.7 kg) were analysed using a non-probability sampling technique by convenience. Considering their specialty distance, in this sample we have 33.3% sprinters, 51.5% middle-distance swimmers, and 15.2% long-distance swimmers. Since these are young age groups, this circumstance could change as the athlete's development progresses. Participants joined the study voluntarily after being informed about the research objectives and their parents signed the informed consent form. The program was presented to 5 training groups that met the conditions of having national competition and a minimum training volume. The initial measurements were carried out between September 15 and 24, 2023 and the final report in July 2024. The guidelines for ethical research in humans set out in the 2013 revision of the Declaration of Helsinki were followed. This study was authorised by the corresponding Research Committee under code no. 23/415-E, for the development of a multi-institution research project in 2023. The sample was selected based on convenience and accessibility. Although the sample size was relatively small, which may limit the statistical power of the analyses, this limitation was primarily due to the accessibility of the entire target group and its coaching staff, as well as the logistical feasibility of conducting the study under real-world conditions with the athlete and technical team. The material resources available for the research, and the way they were utilized, determined the timelines for accessing the sample. Following the proposal to several organizations, only one agreed to provide the necessary resources to carry out the research, while the others declined. Consequently, the study adopts an exploratory approach, which should be considered when interpreting the results. Replicating the study with a larger number of athletes would be both appropriate and of considerable interest.

### Instruments

The variables presented were selected and the data collection techniques were used, in agreement with teams of related researchers in other sports so that they could provide the opportunity to relate data, understanding the specificity and particularity of swimming compared to the rest [29]. In this study, only data relating to the objectives set out will be presented.

The variables and instruments for their collection were

*Sociometric data*: sex, full date of birth (quartile=Q-age, day, month and year) [29], training data (number of sessions and volume in time spent [minutes/week dry and water]) via self-administered questionnaires.

*Body composition data and anthropometric values*: height (cm) via precision stadiometer with 0.001 m scale (SECA GmbH & Co. KG, Hamburg, Germany); weight (kg) and BMI (body mass index) via Inbody 770 bioimpedance platform (InBody Co., Ltd., Seoul, Korea).

*Academic Performance (AP)*: by means of the final average grade of the course the swimmer was in, after the swimmers provided a copy of their official final grades to the research team.

*Final sports performance level obtained in the season (SP)*: final classification at territorial and national level, ratified through the results published on the official websites of the Madrid Swimming Federation and the Royal Spanish Swimming Federation.

*Coach's subjective perception of the athlete's daily performance (Coach Perception)*: ad hoc questionnaire with 5 items, evaluated from 1 to 5 in relation to each swimmer's assessment of the following statements, with 1 being the lowest value (completely disagree) and 5 the highest score (completely agree):

1. *- Understands the contents and explanations given by the trainer in the training sessions.*

2. *- Carries out the proposed tasks in an appropriate manner according to the instructions given.*

3. *- Modifies their behaviour and changes their execution in relation to the possible corrections and FB provided by the coach.*

4. *- Shows maturity by enabling their proper incorporation into their training group.*

5. *- Possesses tools to compensate for deficits in some facet of training and applies them.*

This instrument has not been previously validated by any study; therefore, the data it provides are exploratory and complementary to the main research. These data should be interpreted with caution and represent a known limitation; proposing the validation of this tool with a larger sample in future studies is considered worthwhile. However, a statistical reliability test of this tool was conducted, yielding acceptable results (Cronbach's alpha = .761; McDonald's ω = .792).

### Design and procedures

Quantitative research [3], of descriptive subtype or approach (collection, analysis and presentation of data through quantitative measures), correlational and explanatory, was conducted through direct and indirect data collection. The aim is to analyse variables related to RAE in this sample of young swimmers. The initial research hypothesis, based on previous literature, was that RAE could influence sports performance and some of the variables that may affect it.

The project was presented to all swimmers included in the 5 oldest training groups of the same aquatic institution, which have national reference competitions and both type of training sessions: water and dryland. The inclusion criteria were: swimmers with a valid federation license; belonging to training groups with at least 5 sessions per week and at least 3 years of experience in competitive-level; those who completed an informed consent form signed by them and their legal guardians after receiving information about the protocol and objectives of the study. Regarding the exclusion criteria, swimmers who did not comply with regular training in their groups (attendance <85%); those athletes who did not attend any of the measurements or who had an injury that prevented them from performing the planned physical tests were considered excluded.

After developing the research design and obtaining authorisation from the research committee, data collection was planned for a week that did not interfere with the training loads of the first cycle and quarter of the season, on a day with no previous training load. On the measurement days, several designated areas were set up to allow rotating groups to complete the tests and provide the required data. Consents and authorisations were collected from the participants to form the study group. The previous data of each swimmer were taken, as well as the rest of the personal data. Body composition and anthropometric data were collected with the swimmers wearing swimsuits, barefoot, and dry. Measurements were conducted following standardized procedures using a stadiometer and the instructions provided by the InBody device. Finally, at the end of the last competitive cycle, the coaches' impressions of the swimmers belonging to their group were collected and the most relevant results of each swimmer's season were obtained from the official results of the

federations. In addition, the swimmers provided copies of their final report cards, and the information collected was kept strictly confidential.

Data collection was carried out by a research team composed of four PhD holders in Physical Activity and Sports Sciences, each with more than ten years of teaching experience and expertise in the areas of sports training, education, health, and movement structure. In addition, two of them had extensive experience in the field of aquatic activities. The study also had the support of the organization's technical sports staff—five coaches (four senior coaches and one assistant coach) with substantial practical experience in training groups, ranging from eight years (the least experienced) to twenty-five years (the most experienced).

## Data analysis

First, the database was cleaned and the Mahalanobis distance was calculated to check for outliers. Descriptive statistics, means, standard deviations, skewness and kurtosis values were calculated for the variables under study. The normality of the variables was checked using the Shapiro-Wilk and Levene tests to verify the homoscedasticity of the variances. A one-way ANCOVA was performed to assess the effect of quarter (RAE) of birth on sport level and academic grade point average. ANCOVA effect sizes were expressed with omega squared ($\omega 2$), considering values of <0.06 (small), 0.06 to 0.014 (medium), and >0.014 (large). Cohen's d effect size (ES) was calculated using the formula $ES = t/\sqrt{(n)}$ [30]. The interpretation of ES was 0.2 (irrelevant), 0.2 to 0.6 (small), 0.6 to 1.2 (moderate), 1.2 to 2.0 (large), and >2.0 (very large) [31]. Subsequently, to examine the relationship between the variables, Pearson's correlation test was performed. Correlations were interpreted as small (0.10–0.29), moderate (0.30–0.49), large (0.50–0.69), very large (0.70–0.89) or extremely large (≥ 0.90) [32]. A 95% confidence interval (CI) was considered, and the significance level was $p* < .05$, $p** < .001$. All analyses were run with the statistical packages IBM SPSS 25.0 and Jamovi 2.6.22 version.

## Results

### Descriptive analysis

The mean, median, standard deviation, distribution means (skewness, kurtosis) and homogeneity values (*Shapiro Wilk*) of the variables under study were calculated.

Table 1 shows the descriptive statistics for the demographic variables, variables related to year of birth, days of training, anthropometric variables and coach perception variables about the behaviour of the swimmers during training (Coach perception 1, 2, 3, 4 and 5).

The results obtained in the averages show that the sample has a relative age tending towards the second part of the year, eminently female and with a high number of training days. The weight and BMI values are within the average healthy population values and within what is to be expected in athletes. The average sporting level reached at the end of the season is in the regional top 10 and the average mark is slightly above a "B" (7/10). As for the coach's perception of daily performance, all the items exceed the average values with a positive perception in this sample, with the highest values in items 3 and 4, related to the swimmer's ability to adapt and his or her maturity to integrate into the activity.

The descriptive values of the variables by birth quartiles (Q-age) (view Table 2) show a consistent trend suggesting the possible influence of the effect of Relative Age (RAE) on the sporting and academic performance of the swimmers. Swimmers born in the earliest quartiles (Q1 and Q2) have generally better sporting and academic results, as well as anthropometric profiles that could favour performance, such as a lower BMI and higher body mass ratio adjusted for the type of event.

In contrast, Q4 swimmers show the lowest sporting performance and the greatest dispersion in academic performance. Although these differences do not always reach statistical significance due to the sample size, they reinforce the idea that RAE may act in interaction with other variables such as anthropometric profile or sport maturity, potentially affecting the competitive and academic development of young athletes.

**Table 1. Statistical descriptive objective variables about participants.**

| Variables | M | Me | SD | A | C | Shapiro Wilk |
|---|---|---|---|---|---|---|
| Q-age | 2.55 | 3 | 1.25 | −0.160 | −1.645 | 0.804 |
| Category | 1.58 | 1 | 0.83 | 1.313 | 0.937 | 0.714 |
| Sex | 1.73 | 2 | 0.45 | −1.069 | −0.915 | 0.558 |
| Training-days | 5.88 | 6 | 0.54 | −0.098 | 0.501 | 0.718 |
| Weight | 51.47 | 50.5 | 9.28 | 0.271 | −0.417 | 0.983 |
| BMI | 20.55 | 20.4 | 2.56 | 0.613 | 0.689 | 0.969 |
| Sport Performance level | 1.94 | 2 | 0.7 | 0.790 | −0.253 | 0.822 |
| Mean qualification | 2.21 | 2 | 0.60 | −0.099 | −0.284 | 0.758 |
| Coach Perception 1 | 2.58 | 3 | 0.66 | −1.323 | 0.619 | 0.656 |
| Coach Perception 2 | 2.55 | 3 | 0.62 | −1.032 | 0.140 | 0.695 |
| Coach Perception 3 | 2.85 | 3 | 0.36 | −2.037 | 2.287 | 0.431 |
| Coach Perception 4 | 2.73 | 3 | 0.57 | −2.057 | 3.413 | 0.531 |
| Coach Perception 5 | 2.48 | 3 | 0.57 | −0.488 | −0.769 | 0.710 |

Q-age (RAE, relative age); *M,* mean; *Me,* median; *SD,* statistical deviation; A, skewness; C, kurtosis; W, normality statistic.

**Table 2. Descriptive and quartile distribution data in relation to the variables under study.**

| Variables | Q-age | N | Mean | Medium | SD | Skewness | Kurtosis | EE | W | p |
|---|---|---|---|---|---|---|---|---|---|---|
| Sport performance | 1 | 11 | 2.18 | 2 | 1.07 | 0.73 | −0.35 | 1.28 | 0.84 | 0.03 |
| | 2 | 3 | 2.67 | 3 | 1.52 | −0.93 | NaN | Inf | 0.96 | 0.63 |
| | 3 | 9 | 1.78 | 2 | 0.66 | 0.25 | −0.04 | 1.40 | 0.81 | 0.02 |
| | 4 | 10 | 1.60 | 1.00 | 0.84 | 1.00 | −0.66 | 1.33 | 0.71 | 0.00 |
| Academic Perfomance | 1 | 11 | 8.23 | 8.00 | 1.07 | 0.20 | −1.26 | 1.28 | 0.93 | 0.46 |
| | 2 | 3 | 8.67 | 8.40 | 1.12 | 1.01 | NaN | Inf | 0.96 | 0.60 |
| | 3 | 9 | 8.21 | 8.30 | 0.74 | 0.39 | −0.56 | 1.40 | 0.97 | 0.90 |
| | 4 | 10 | 7.97 | 7.90 | 1.26 | −0.20 | −1.20 | 1.33 | 0.94 | 0.55 |
| BMI | 1 | 11 | 20.52 | 19.80 | 3.01 | 1.42 | 3.26 | 1.28 | 0.89 | 0.15 |
| | 2 | 3 | 21.70 | 20.41 | 2.24 | 1.73 | NaN | Inf | 0.75 | 0.00 |
| | 3 | 9 | 20.33 | 20.00 | 2.57 | 0.45 | −1.24 | 1.40 | 0.91 | 0.35 |
| | 4 | 10 | 20.43 | 21.30 | 2.38 | −0.49 | −0.95 | 1.33 | 0.94 | 0.54 |
| Weight | 1 | 11 | 48.95 | 45.40 | 11.40 | 1.34 | 0.84 | 1.28 | 0.81 | 0.01 |
| | 2 | 3 | 55.63 | 54.80 | 3.23 | 1.08 | NaN | Inf | 0.95 | 0.57 |
| | 3 | 9 | 51.19 | 50.50 | 7.46 | −0.15 | −0.04 | 1.40 | 0.96 | 0.89 |
| | 4 | 10 | 53.24 | 55.00 | 9.64 | −0.61 | 0.09 | 1.33 | 0.96 | 0.80 |
| Height | 1 | 11 | 1.54 | 1.54 | 0.09 | 0.31 | 0.54 | 1.28 | 0.97 | 0.95 |
| | 2 | 3 | 1.60 | 1.61 | 0.04 | −0.72 | NaN | Inf | 0.98 | 0.72 |
| | 3 | 9 | 1.59 | 1.59 | 0.09 | −0.30 | −1.48 | 1.40 | 0.92 | 0.41 |
| | 4 | 10 | 1.61 | 1.58 | 0.10 | 1.16 | 0.61 | 1.33 | 0.88 | 0.15 |

Q-age (RAE, relative age); *M,* mean; *Me,* median; *SD,* statistical deviation; A, skewness; C, kurtosis; W, normality statistic: Shapiro-Wilk.

Regarding the correlation observed between the variables under study, the positive and statistically significant associations shown in Table 3 are described below in annexes. The academic performance average correlates with sport performance with the perception of coach 1 and coach 4. The perceptions of the coaches correlated positively with each other,

**Table 3. Bivariate correlation analysis of the variables under study.**

| | APA | SP | C1 | C2 | C3 | C4 | C5 | TS | Sex | Cat | Month | RAE | BMI | Weight |
|---|---|---|---|---|---|---|---|---|---|---|---|---|---|---|
| APA | — | | | | | | | | | | | | | |
| SP | 0.368* | — | | | | | | | | | | | | |
| C1 | 0.405** | 0.105 | — | | | | | | | | | | | |
| C2 | 0.128 | 0.431** | 0.335* | — | | | | | | | | | | |
| C3 | 0.179 | 0.236 | 0.406** | 0.598*** | — | | | | | | | | | |
| C4 | 0.316* | 0.167 | 0.293* | 0.229 | 0.212 | — | | | | | | | | |
| C5 | 0.276 | 0.285 | 0.461** | 0.618*** | 0.324* | 0.428** | — | | | | | | | |
| TS | −0.040 | −0.129 | 0.043 | 0.017 | 0.055 | −0.134 | −0.023 | — | | | | | | |
| Sex | −0.017 | 0.114 | −0.168 | −0.174 | −0.069 | −0.005 | −0.261 | −0.009 | — | | | | | |
| Cat | 0.016 | −0.172 | 0.218 | 0.234 | 0.046 | −0.166 | 0.218 | 0.573*** | 0.086 | — | | | | |
| Month | −0.079 | −0.249 | 0.147 | 0.012 | −0.121 | 0.089 | 0.267 | 0.509** | −0.187 | 0.462** | — | | | |
| RAE | −0.113 | −0.265 | 0.132 | 0.024 | −0.060 | 0.095 | 0.280 | 0.426** | −0.205 | 0.445** | 0.964*** | — | | |
| BMI | −0.077 | −0.430 | 0.249 | −0.242 | −0.284 | −0.022 | −0.079 | 0.009 | −0.132 | 0.223 | 0.108 | 0.036 | — | |
| Weight | −0.012 | −0.211 | 0.236 | −0.069 | −0.328 | −0.064 | 0.009 | 0.376* | −0.143 | 0.548*** | 0.308* | 0.218 | 0.718*** | — |

APA, Academic Perfomance Average; SP, Sport Performance; C1, Coach-1; C2, Coach-2; C3, Coach-3; C4, Coach-4; C5, Coach-5; TS, Training sessions; Cat, category; RAE, relative age effect; BMI, body mass index; $H_a$ = positive correlation; * $p < .05$, ** $p < .01$, *** $p < .001$.

except for coach 4 which only correlated positively with the perception of coach 1. Finally, the category correlated positively with the training sessions. Positive and significant correlation (through Pearson's *p* and Spearman's *s*) is observed between all the variables.

Finally, frequency data are provided, highlighting the athlete profile in terms of frequency of Q-age, sporting level and average score (Table 4).

In the present sample it can be observed that the month of birth of 58% of the sample would be located in the second half of the year. In terms of sporting level, it can be observed that 60% have a high regional level and 25% are able to go to national competitions. We can highlight that the great majority of the swimmers analysed (90%) have an academic performance that ranges from a "B" to an "A" in the final grade of their studies.

## Multivariate analysis MANCOVA (covariance)

The analysis of covariance allowed us to analyse the relationship between Q-age, month of birth and days of training (as independent variables) with sport level, body weight, BMI and academic grade point average (dependent variables), taking into account the influence of age and swimming category (covariates). The results obtained showed statistically significant differences between Q-age (Wilk's $\Lambda = .16$, F (12.59), $p = .001$), month of birth (Wilk's $\Lambda = .16$, F (14.17) = 1, $p = .001$), number of training days (Wilk's $\Lambda = .16$, F (6.97), $p = .003$) and body weight mediated by sport category (covariate).

## Multivariate ANCOVA analysis (covariance)

ANCOVA analysis reported the statistically significant relationship of RAE on sport performance ($F = 5.01$; $p = .036$) when BMI is taken as a covariate with a moderate effect size ($\eta^2 = .14$; $\omega^2 = .10$). Likewise, RAE showed a statistically significant relationship with mean grade ($F = 0.433$; $p = .027$; $\eta^2 = .14$; $\omega^2 = .11$), year of birth ($F = 0.143$; $p = .038$; $\eta^2 = .12$; $\omega^2 = .095$) and month of birth ($F = 2.293$; $p = .027$; $\eta^2 = .063$; $\omega^2 = .033$) when swimming category is set as a covariate.

Furthermore, the perception of the coach ("Coach perception": 1, 2, 3, 4 and 5, independent variable with several levels) was related to the sport performance level (dependent variable). Only "Coach perception 2" (*performs the tasks*

**Table 4. Frequency of age quartiles, level of sport achieved and mean academic grades achieved.**

| Q-age | Frequencies | % of Total | Accumulated % Cumulative |
|---|---|---|---|
| 1 | 11 | 33.3% | 33.3% |
| 2 | 3 | 9.1% | 42.4% |
| 3 | 9 | 27.3% | 69.7% |
| 4 | 10 | 30.3% | 100.0% |
| **Sport Performance** | | | |
| 1 | 13 | 39.4% | 39.4% |
| 2 | 12 | 36.4% | 75.8% |
| 3 | 5 | 15.2% | 90.9% |
| 4 | 3 | 9.1% | 100.0% |
| **Average mark** | | | |
| 1 | 3 | 9.1% | 9.1% |
| 2 | 20 | 60.6% | 69.7% |
| 3 | 10 | 30.3% | 100.0% |

Q1 (January to March); Q2 (April to June); Q3 (July to September); Q4 (November to December); Sport Performance (1: Regional; 2: top 10 Regional; 3: National; 4: top 10 National); Average grade (1: good; 2: remarkable; 3: outstanding).

*adequately in relation to the instructions provided by the coach*) showed statistically significant differences ($F = 1.198$; $p = .033$; $\eta^2 = .055$; $\omega^2 = .040$) on sport performance, when the category of play is set as a covariate.

In order to further explore the significant differences identified through ANCOVA, post hoc analyses using Tukey's HSD were conducted (view Table 5). These comparisons revealed meaningful trends between quartiles of birth. In particular, moderate to large effect sizes were observed in sport performance differences between Q2 and Q4 (Cohen's $d = 0.73$) and Q1 and Q4 ($d = 0.64$), suggesting that relatively younger athletes may experience a disadvantage compared to their older peers.

These trends support the statistically significant ANCOVA result that linked RAE to sport performance when BMI was included as a covariate ($F = 5.01$; $p = .036$; $\eta^2 = .14$; $\omega^2 = .10$). Although the post hoc comparisons did not reach conventional significance thresholds after multiple comparison correction, the magnitude of the differences reinforces the relevance of RAE in this sample. Additionally, the significant effect of month of birth ($F = 2.293$; $p = .027$; $\omega^2 = .033$) aligns with these findings, as the largest disparities were found between swimmers born early (Q1) and late (Q4) in the calendar year.

**Table 5. Summary of ANCOVA and Post Hoc Results.**

| Dependent Variable | Factor/ Covariate | F | p | η²/ ω² | Post Hoc (Tukey HSD) Summary |
|---|---|---|---|---|---|
| Sport Performance | Q-age (BMI as covariate) | 5.01 | 0.04 | η² = .14; ω² = .10 | Q2–Q4: d = 0.73; Q1–Q4: d = 0.64 (ns) |
| Academic Performance | Q-age (Category as covariate) | 0.43 | 0.02 | η² = .14; ω² = .11 | No pairwise significance; small n |
| Year of Birth | Q-age (Category as covariate) | 0.14 | 0.04 | η² = .12; ω² = .095 | Not applicable |
| Month of Birth | Q-age (Category as covariate) | 2.29 | 0.03 | η² = .063; ω² = .033 | Q1–Q4: d = −11.28***; Q1–Q3: d = −7.64*** |
| Sport Performance | Coach Perception 2 (Category as covariate) | 1.19 | 0.03 | η² = .055; ω² = .040 | CP2–3 vs CP2–5: d = −2.56*; CP2–2 vs CP2–5: d = −1.72* |

*p* (significant value); η²/ ω² (effect size).

Taken together, these results suggest that the influence of RAE on sport performance is conditional and may be mediated by physical maturity indicators such as BMI, as well as reinforced by contextual factors like the coach's perception, which also demonstrated significant effects (Coach 2: F = 1.198; p = .033; ω² = .040).

This Table (5) summarizes the main ANCOVA results and post hoc comparisons related to the effect of RAE on sport performance, academic performance, and related variables, including effect sizes and pairwise differences between quartiles where relevant.

## Discussion

This study examined the relationship between swimmers' relative age and their sporting performance at the end of the season. Additionally, the influence of relative age on anthropometric profile, training conditions and coaches' subjective perception on sporting and academic performance was examined. While relative age did not have a direct effect on performance, it did have an influence when factors such as month of birth, BMI, weight and sports category were considered. Furthermore, coaches' perceptions were related to the competitive category.

### The relationship between RAE and sports performance

The present study shows no direct relationship between RAE and the rest of the variables in a significant way. The initial hypothesis was not confirmed by this research. Previous literature on swimming indicates that early maturation, linked to the relative age effect (RAE), provides physical advantages such as increased size, strength, endurance and speed. This can lead to differences in performance within the same competitive category [8,33,34]. However, the data from this study showed no significant direct differences between the two variables. This is inconsistent with other findings involving swimmers, which did reveal differences in subjects' physical condition and performance in specific tests [8,35–37]. There is a possibility that the sample size used in the present study could also be a limiting factor in not finding significant differences. In this regard, it can also be indicated as a limitation that the sample is not gender-balanced, as the accessibility of the sample provided a greater number of female swimmers. Nevertheless, no differentiation has been observed in relation to gender and RAE on sport performance. Although the results found may contradict other research where there is differentiation between genders in their performance reach and RAE [11,37,38]. Differentiation also appears within the same female gender across quartiles when discussing competition levels and sports results. Greater differences are established in preadolescent and adolescent ages, where differences in the degree of physical maturation can be striking and gain importance, especially in disciplines with high physiological demands [39], but these disparities can be observed throughout the entire span of competitive age groups [40]

Even so no such direct relationship has been found, it has been observed that more than half of the swimmers who achieve the highest sporting results belong to the first two trimesters of the year. This fact is directly related to the findings of other studies, in which an over-representation of swimmers born in the first two quarters of the year [33,41] in the most referential positions is observed. Despite that, these studies also fail to obtain a significant relationship when comparing between groups of swimmers born in the first two trimesters of the year. Only when comparing swimmers born in the first two quarters of the year with those born in the last two quarters of the year is it observed that the RAE has a greater representation and effect on sporting performance.

On numerous occasions, physical maturity and cognitive ability [42,43] are attributed to the RAE as the main enhancers in sports performance, without considering the talent and skill of the athletes, as well as the so-called social agents. Hancock et al. [44] argue that these social agents have a greater influence on the RAE than physical and cognitive maturity itself, in that depending on how the RAE is interpreted by parents, coaches, and the athletes themselves, performance outcomes will differ and greater inequalities will be generated based on this, regardless of physical development. This fact could justify the overrepresentation of swimmers from the first two quarters among the top-ranked in competitions.

Although other sport disciplines have found a direct effect of RAE on sport performance [45] and the selection of athletes for major events as part of representative teams [46], it should not be forgotten that much of the research has been based on team sports [9], with less focus on individual sports and those with higher technical demands such as swimming [39,47]. For this reason, more research is needed in swimming to determine whether the approach in this sport could be different from that given in other sports when considering the RAE as a predictor of performance. Perhaps the mechanisms of self-control, information management, and compensation tools of swimmers in relation to training task outcomes in the specific aquatic context could be modified between these athletes and those from other disciplines [11].

In relation to individual swimmer characteristics, another consideration that would support the results obtained in the study is the possibility that athletes who are relatively "younger" because they were born in the last quartiles of the year, present equivalent or superior technical and psychological skills. In the sport of swimming, the importance of the impact of technical ability in reducing forward resistance on sport performance is well known [14,48,49]. This trade-off of technical aspects against this chronological difference between swimmers would lead us to think that RAE alone might not be a fully effective predictor of performance in swimmers. Several studies have pointed out that the effect of relative age remains unclear in disciplines where technical skills must be applied in open and uncertain contexts, such as combat sports [50], or in those involving equipment use or participation in intradisciplinary specialties, as seen in gymnastics [51,52].

### The relationship of the RAE with other influential variables

A substantial portion of the literature examining the REA within individual sports disciplines concentrates on identifying the athlete's birth quartile in relation to their athletic performance. These studies consistently confirm the presence of the RAE, demonstrating that athletes born earlier in the selection year often exhibit performance advantages. However, the strength of these associations tends to diminish when additional variables are considered—such as anthropometric characteristics, technical proficiency, and daily engagement in the training process [53]. In this study we also evaluated, as secondary objectives, the relationship of the RAE on the anthropometric profile, the training conditions and the subjective perception of the coaches on the daily performance of the swimmers; as well as the relationship of these three variables with sports performance (SP) and academic performance (AP). As mentioned above, no direct relationship was found between these variables.

The results of the covariate analysis showed an influence between RAE and SP, both only when the competition category, year and month of birth are established as covariates. Being born shortly before the cut-off date used to establish sport competition categories or academic years may in many cases cause inequality of opportunity among the components of that year or category, as a matter of maturational development and/or experience [1,49].

Incorporating the intervention of body weight and SP, in the results obtained by Strzala and Tyka [54], sprint swimmers have a higher body weight and lean mass than endurance swimmers. This data highlights the need to consider the distance of the event when taking into account the effect of the swimmer's body weight as an indicator of performance. In addition, another important factor to consider when establishing this relationship is the somatotype structure of the swimmer [55]. In fact, it is possible to find some statistically significant negative correlations between swimmers' performance and mesomorphic variables [56]. This finding places the somatotype of the swimmer as another important performance indicator for the different distances of the competition.

Along with the above, the results indicated that RAE has a statistically significant influence on SP when BMI is taken as a covariate. A lower BMI in swimmers can positively influence swimming endurance and power, as well as propulsive power and even buoyancy. In this study, the results indicate that swimmers born in the first quartiles of the year have a lower BMI, and therefore more favourable for SP, than those born in the last quartiles, which is in line with the findings of other studies [55].

When presenting the relationship of SP with training days, academic grade obtained and year of birth, together with swimming category, the results obtained in this study, as expected in congruence with the scientific literature

reviewed [1,57], point to a significant and positive relationship between training days and the average academic grade obtained by the swimmers. This means that the higher the number of training days, the higher the average academic grade. This finding is consistent with those obtained in other studies in which regular sport practice has a positive influence on the academic performance of athletes [58]. The ability to concentrate and commitment, coping with stress, resilience and perseverance, among others, are indispensable components present in sport practice that are directly related to what is reflected in the academic environment [57]. All of them make sport an interesting facilitator in academic learning, providing the athlete with the possibility of developing important skills for academic success.

Finally, an attempt was made to check whether the coach's daily observations of his or her athletes could be related to their final performance. This information was initially collected to complement the main data, adding value by providing a more complete view of athletes' daily performance in relation to their final sports outcomes. Since an unvalidated exploratory tool was used, the interpretation of these findings should be approached with caution, as this methodological limitation is acknowledged and may affect the generalizability and robustness of the conclusions drawn. The results did show a significant relationship between the coach's perception and the SP when attending to the performance of the tasks in an adequate way when they adapt to the instructions provided by the coach. This could mean that swimmers who perform tasks adequately following instructions provided by their coach on a regular basis are those who show a higher level of sport performance participating in higher level competitions [2]. Swimmers who compete at higher levels have more experience of training and competition time, which at the same time gives them a higher level of physical and cognitive maturity. This may also be true for coaches and their professional predisposition [59]. Furthermore, the results emphasise the importance of effective communication between coaches and swimmers, as well as the swimmers' ability to understand and apply cues. This ability hinges on the athlete's level of maturity, as well as the coach's ability to convey clarity, confidence, motivation and leadership [60]. The role of a coach influences not only strictly sporting aspects (physical, technical and tactical development) but also psychological aspects (motivational, emotional...) [61].

In addition, there was a moderate trend, though not a significant one, suggesting that heavier swimmers are more likely to adjust their behaviour and improve their technique in response to feedback and cues from their coach. The possible interpretation of these results lies in the fact that swimmers with a higher body weight are those with an advanced physical and cognitive growth due to their chronological age (older group), which gives them a higher level of maturity and experience, compared to younger swimmers, and allows them to understand and apply more easily the indications given by the coach, although they could also fall into more disruptive behaviours [27], which could also be influenced by their interpersonal style of intervention on their personal needs [62,63].

It is understood that further studies with a larger sample of participants—one of the main limitations of the present study—are necessary to corroborate the influence of RAE on different swimming styles or race distances. This issue is acknowledged in the current research and may limit the generalizability and extrapolation of the findings, which should therefore be interpreted with caution. Additionally, further investigation is warranted to explore the potential implications of RAE across other age groups and to compare results in individual sports beyond swimming. Considering aspects related to the coach's subjective assessment in the athlete's daily routine, this information can broaden the overall understanding of the training process and its contribution to final performance, but only as a guideline. The proposed tool, when applied to a larger number of athletes and training groups, may attain appropriate levels of reliability and validity, incorporating relevant power data following future studies.

In this regard, tailoring training programs based on evidence of the physiological demands and progressive developmental characteristics of swimmers—alongside a more holistic perspective from coaches regarding training group management—could help introduce compensatory strategies in planning. Such an approach would aim to foster more personalized performance outcomes.

## Conclusions

The findings of this study lead us to conclude that, within the analyzed sample, the Relative Age Effect (RAE) did not exert a direct influence on sports performance (SP) or on the other variables examined. This outcome contrasts with previous literature, which has identified such an effect. However, when body composition is considered as a covariate, a relationship emerges between RAE and SP, along with notable associations with academic performance (AP) and the number of training days. These results suggest that an athlete's capacity for adaptation and their consistent effort to improve—perceived through the lens of the coach—play a significant role in determining final athletic performance.

While RAE continues to appear as a potential influencing factor in swimmers' performance, its impact is mediated by various contextual elements surrounding daily training. These include training conditions (such as facilities, equipment, scheduling, and nutrition), the coach's approach to planning technical, physical, and psychological development, and the athlete's own motivation and engagement. Therefore, the implementation of a comprehensive and well-structured training plan may enhance the feasibility of achieving performance goals and support progression toward higher levels of competition.

## Supporting information

**S1 Table. Reliability Analysis.**
(DOCX)

**S2 Table. Coach perception questionnarie.**
(DOCX)

## Acknowledgments

This study has counted with the collaboration of the Rivas Swimming Sport Association, thanks to the collaboration and knowledge transfer agreement between the Aqualab research group of the Universidad Europea de Madrid and the Madrid Swimming Federation. The authors thank all the participating athletes, their families and the sports club for their altruistic participation in the study. They also thank the Universidad Europea de Madrid for providing the facilities and resources for carrying out the research.

## Author contributions

**Conceptualization:** Mendoza-Castejón, D, Trinidad A, De la Calle, L.M, Belando-Pedreño, N.

**Data curation:** Mendoza-Castejón, D, Belando-Pedreño, N.

**Formal analysis:** Mendoza-Castejón, D, Trinidad A, De la Calle, L.M, Belando-Pedreño, N.

**Investigation:** Mendoza-Castejón, D, Trinidad A, Belando-Pedreño, N.

**Methodology:** Mendoza-Castejón, D, De la Calle, L.M, Belando-Pedreño, N.

**Project administration:** Mendoza-Castejón, D, Trinidad A.

**Resources:** Mendoza-Castejón, D, Belando-Pedreño, N.

**Software:** Belando-Pedreño, N.

**Supervision:** Mendoza-Castejón, D, Trinidad A.

**Validation:** Mendoza-Castejón, D, De la Calle, L.M, Belando-Pedreño, N.

**Visualization:** Mendoza-Castejón, D, Trinidad A, De la Calle, L.M, Belando-Pedreño, N.

**Writing – original draft:** Mendoza-Castejón, D, Trinidad A, De la Calle, L.M, Belando-Pedreño, N.

**Writing – review & editing:** Mendoza-Castejón, D, Trinidad A, De la Calle, L.M, Belando-Pedreño, N.

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
