## [Decision Letter · Decision Letter 0]

16 Jun 2025

Dear Dr. Trinidad Morales,

Please submit your revised manuscript within Jul 31 2025 11:59PM, from the date of this report. If you will need more time than this to complete your revisions, please reply to this message or contact the journal office at plosone@plos.org . A rebuttal letter that responds to each point raised by the academic editor and reviewer(s). You should upload this letter as a separate file labeled 'Response to Reviewers'.A marked-up copy of your manuscript that highlights changes made to the original version. You should upload this as a separate file labeled 'Revised Manuscript with Track Changes'.An unmarked version of your revised paper without tracked changes. You should upload this as a separate file labeled 'Manuscript'.

We look forward to receiving your revised manuscript.

Kind regards,

Mohamed Ahmed Said, Ph.D.

Academic Editor

PLOS ONE

Journal Requirements:

2. Please ensure that you have specified:

a) Did participants provide their written or verbal informed consent to participate in this study?

b) If consent was verbal, please explain i) why written consent was not obtained, ii) how you documented participant consent, and iii) whether the ethics committees/IRB approved this consent procedure."

- In consent please state in Ethics Method section and manuscript if it is written or verbal. If consent was verbal, please explain a) why written consent was not obtained, b) how you documented participant consent, and c) whether the ethics committees/IRB approved this consent procedure.

Reviewers' comments:

Reviewer's Responses to Questions

**Comments to the Author**

1. Is the manuscript technically sound, and do the data support the conclusions?

Reviewer #1: Yes

Reviewer #2: Partly

Reviewer #3: Partly

2. Has the statistical analysis been performed appropriately and rigorously?

Reviewer #1: I Don't Know

Reviewer #2: Yes

Reviewer #3: Yes

3. Have the authors made all data underlying the findings in their manuscript fully available?

Reviewer #1: Yes

Reviewer #2: Yes

Reviewer #3: Yes

4. Is the manuscript presented in an intelligible fashion and written in standard English?

Reviewer #1: Yes

Reviewer #2: Yes

Reviewer #3: Yes

Reviewer #1: Overall, the study addresses an important and timely topic—the influence of the relative age effect (RAE) on the athletic and academic performance of young swimmers—using a generally sound methodological approach. The article is also fairly well written, with clear language and a logical structure that makes it accessible to readers. However, the study as a whole comes across as somewhat unfocused and overly broad.

The inclusion of a wide range of variables—from anthropometric data and academic grades to the coach’s subjective evaluation—combined with a relatively small and heterogeneous sample, makes it difficult to draw clear conclusions and somewhat blurs the central message of the article. A more focused design, with fewer but more coherent variables, and a larger, more homogeneous sample—e.g., grouped by distance specialization or age category—would likely result in a clearer and more impactful contribution.

Below, I present specific comments and suggestions related to section of the manuscript.

1) Introduction

Although the introduction provides a broad and comprehensive overview of the literature on RAE in sports, it lacks a clearly defined research gap that would justify this study. The authors cite numerous relevant studies and refer to multiple performance-related factors, but they do not clarify what specifically is missing in the current body of knowledge—especially in the context of swimming as a technical and individual discipline. Furthermore, it remains unclear whether the originality of this research lies in the national setting, the age group studied, or the competitive level analyzed. To strengthen the rationale for the study, the authors should more explicitly define the knowledge gap and explain how their study aims to address it.

Additionally, while the authors do state the objectives of the study at the end of the introduction, they fail to formulate specific, testable hypotheses. Given the quantitative nature of the research and its aim to explore relationships between RAE and multiple performance-related variables, the absence of clearly articulated hypotheses weakens both the methodological clarity and the coherence of the interpretation. The authors are encouraged to clearly state their assumptions and expectations, including the predicted direction and strength of the relationships under investigation.

2) Participants

While the sample includes national-level swimmers, the age range (mean age 15 ± 2.08 years) introduces substantial variability that may impact the interpretation of the results. In adolescent athletes, particularly around the age of 15, even one year can correspond to significant differences in biological maturity, anthropometric characteristics, and physical performance. This variation becomes even more pronounced when comparing athletes with different specializations, such as sprinters versus long-distance swimmers, who differ considerably in their developmental and physiological profiles. The presence of athletes with an age difference of more than two years within the same sample means that the study is comparing individuals at very different stages of physical and psychological development and sport specialization, which could limit the validity of the conclusions drawn.

3) Discussion

The discussion begins by restating the main objectives of the study in terms that closely mirror the introduction. While briefly revisiting the aims is acceptable, this repetition adds little value unless it is immediately tied to the study’s findings. It is recommended that the authors revise the opening to more directly present and interpret their key results in light of the study’s goals.

The discussion covers a wide range of topics and successfully links the study's findings to existing research, addressing both statistically significant and non-significant outcomes. The authors appropriately acknowledge several limitations, such as the small sample size and the imbalance in gender and specialization. However, at times the section feels overly long and resembles a literature review, which can dilute the focus on the study’s own results.

It is commendable that the authors attempt to consider the combined influence of social, anthropometric, and psychological factors. Nevertheless, some of their interpretations are speculative and not clearly supported by their own data—for instance, references to the Pygmalion effect or the influence of coaching style. A more distinct separation between data-driven conclusions and those based purely on theoretical or literature-derived explanations would improve clarity.

Furthermore, the discussion would benefit from a more concise and explicit presentation of the practical implications of the findings, especially for coaches, sport program designers, and federations. Suggested directions for future research are relevant, particularly those that call for the inclusion of measures like biological maturity to better control for individual differences in development.

4) Conclusion

The conclusion could be more concise and avoid repeating results already discussed earlier in the manuscript. It would also benefit from a clearer focus on the practical implications for coaches, training programs, and youth sport planning.

The authors’ suggestions for future research are a strength of the section. Highlighting the need for larger samples and comparative studies in other sports and age groups enhances the broader applicability of the findings. However, these more detailed research proposals might be better placed in the final part of the discussion, rather than in the conclusion itself, which should ideally focus on the key takeaways from the study.

Reviewer #2: Dear Authors,

I want to express my gratitude for the opportunity to review this manuscript.

Congratulations on the study. Some improvements are suggested below, with line indications.

L69-70 – Please consider abbreviating “PA”.

L98 – “relative age” – RAE suggested.

L108 – Please describe the inclusion and exclusion criteria. Please describe all available information related to the subjects´ characterization. Some examples include training routines, years of experience, competitive level, and number of weekly training sessions.

L121-188 - Please describe all methodological details, for example, the procedures in detail, preferably with reference support. Another example is the human resources involved – academic background and experience.

L191-206 – Please consider shorter paragraphs to improve readability (8-12 lines suggested). Please make sure all statistical procedures are described.

L216 – Please revise all tables´ content and format, considering the journal template and instructions for authors.

L207-282 – The reformulation of this section is suggested to become more appealing and easier for readers' interpretation. Please revise the text and tables.

L286 – Only “RAE” suggested, previously in full. Please revise all manuscript.

L290-420 - Please consider improving the quality of the discussion section, namely regarding the study rationale, and with the inclusion of more references.

L420 – Please consider indicating the study limitations and suggestions for future research,

L422 - Please consider short and clear take-home messages, if possible, with practical applications/implications.

L456 - Please double-check the references format. For example: Titles in upper and lowercase; Journal in full and abbreviated.

Reviewer #3: General Feedback

The manuscript presents valuable original research on the relative age effect (RAE) in youth swimmers. The study research question is highly relevant, and the methodology is sound, highlighting how the RAE influences performance conditionally, which is a significant contribution. However, the paper can be strengthened by addressing some areas in different sections as suggested.

Abstract

The abstract can be made more concise while still including all essential information, especially about your novel findings (e.g., direct versus mediated effects).

Introduction

• Given your extensive literature review, are there any specific contradictory findings in previous research that your study directly aims to clarify or challenge?

• The introduction could benefit from a clearer articulation of the specific gap in the literature that this study directly aims to fill, beyond generally exploring RAE in swimming.

• The topic and idea are valid. However, the article reads more like a description than a true scientific analysis guided by a clearly defined theoretical framework. The theoretical framework connecting RAE to swimming performance requires enhancement, and it is important to include a comparison with similar studies on RAE.

Figures and tables

Could a graphical representation of key findings, such as the relationship between birth quartile and performance when mediated by BMI, enhance readers' understanding of your results?

Methods

• Could you elaborate on your rationale for using a convenience sample and discuss any potential biases this might introduce?

• Given the acknowledged sample size limitation; (a). what steps did you take during the analysis to account for this (e.g., sensitivity analyses)? (b) Did you conduct any power analysis to determine the necessary sample size to detect expected effects?

• You have given 5 items on the "coach's subjective perception" based on ad hoc questionnaire. Could you provide a copy of the full questionnaire as supplementary material for reproducibility?

• The specific procedures for data collection should be further explained. For example, how were the questionnaires administered, what were the exact anthropometric measurement protocols if they deviate from standard including reference?

• Are there any other specific procedural details that you could expand upon for greater transparency and reproducibility? e.g., order of measurements, instructions given to swimmers during anthropometry etc.

Statistical Analysis and Results

• Could you present a more detailed breakdown/post hoc analysis of the significant differences in ANCOVA. For instance, could you clarify which quartiles demonstrated significant differences, such as between Q1 and Q4, regarding the means and standard deviations of the dependent variables?

• The ANCOVA results are presented with F-values and p-values, but not the specific group means for the factors involved. Could you provide the means and standard deviations for the dependent variables across different/individual quartiles or months of birth, and training days, to better illustrate the significant differences?

• The specific statistical approach such as Tukey's HSD, FDR etc., to address the issue of multiple comparisons given the number of correlations and ANCOVA analyses performed are not explicitly stated. This represents a limitation in your statistical methodology, as multiple comparison corrections would have strengthened the reliability of your findings.

Discussion

• Could the authors expand on how their findings, particularly the conditional influence of RAE, challenge or refine existing theories on RAE in sports?

• Given the stated limitations in the methods, what specific future research directions are most crucial to build upon these findings?

• How might the acknowledged gender imbalance in the sample affect the generalizability of the findings to male swimmers, even if no gender differentiation was observed within the study?

• Could you expound on how these findings compare to RAE studies in other individual sports?

Conclusions

• Could the practical implications of the findings be more explicitly stated for coaches, parents, and sports organizations?

• How might the "global plan" mentioned in the conclusions be structured or implemented based on these findings?

Additional aspects

• It is stated in the manuscript that, "All data are fully available without restriction," implying deposition, but you haven't provided specific repository or accession numbers for the data. This is crucial for transparency and reproducibility as per the journal guidelines.

• STROBE (Strengthening the Reporting of Observational Studies in Epidemiology) would be relevant guidelines for this work. Please ensure full adherence.

**Do you want your identity to be public for this peer review?** For information about this choice, including consent withdrawal, please see our Privacy Policy

Reviewer #1: No

Reviewer #2: **Yes: ** Mário Espada

Reviewer #3: No

---

## [Author Response · Author response to Decision Letter 1]

12 Aug 2025

Response to Reviewers – Full Letter

We sincerely thank the reviewers for their thoughtful comments and suggestions, which have helped us significantly improve the clarity, methodological rigor, and presentation of our manuscript. Below, we provide detailed, point-by-point responses to each comment.

Response to Reviewers

Reviewer #1: Overall, the study addresses an important and timely topic—the influence of the relative age effect (RAE) on the athletic and academic performance of young swimmers—using a generally sound methodological approach. The article is also fairly well written, with clear language and a logical structure that makes it accessible to readers. However, the study as a whole comes across as somewhat unfocused and overly broad.

The inclusion of a wide range of variables—from anthropometric data and academic grades to the coach’s subjective evaluation—combined with a relatively small and heterogeneous sample, makes it difficult to draw clear conclusions and somewhat blurs the central message of the article. A more focused design, with fewer but more coherent variables, and a larger, more homogeneous sample—e.g., grouped by distance specialization or age category—would likely result in a clearer and more impactful contribution.

Below, I present specific comments and suggestions related to section of the manuscript.

Thank you very much for your comments. We would now like to address some of your observations and specify the changes that have been made to the document accordingly.

1) Introduction

Although the introduction provides a broad and comprehensive overview of the literature on RAE in sports, it lacks a clearly defined research gap that would justify this study. The authors cite numerous relevant studies and refer to multiple performance-related factors, but they do not clarify what specifically is missing in the current body of knowledge—especially in the context of swimming as a technical and individual discipline. Furthermore, it remains unclear whether the originality of this research lies in the national setting, the age group studied, or the competitive level analyzed. To strengthen the rationale for the study, the authors should more explicitly define the knowledge gap and explain how their study aims to address it.

Additionally, while the authors do state the objectives of the study at the end of the introduction, they fail to formulate specific, testable hypotheses. Given the quantitative nature of the research and its aim to explore relationships between RAE and multiple performance-related variables, the absence of clearly articulated hypotheses weakens both the methodological clarity and the coherence of the interpretation. The authors are encouraged to clearly state their assumptions and expectations, including the predicted direction and strength of the relationships under investigation.

We have revised the introduction to more explicitly state the research gap regarding RAE in swimming within the Spanish context. Additionally, we now include clear, directional hypotheses based on the reviewed literature.

2) Participants

While the sample includes national-level swimmers, the age range (mean age 15 ± 2.08 years) introduces substantial variability that may impact the interpretation of the results. In adolescent athletes, particularly around the age of 15, even one year can correspond to significant differences in biological maturity, anthropometric characteristics, and physical performance. This variation becomes even more pronounced when comparing athletes with different specializations, such as sprinters versus long-distance swimmers, who differ considerably in their developmental and physiological profiles. The presence of athletes with an age difference of more than two years within the same sample means that the study is comparing individuals at very different stages of physical and psychological development and sport specialization, which could limit the validity of the conclusions drawn.

The differentiation that may appear among adolescent athletes is well known, and this degree of variability is generally acknowledged. The aim of this study was to establish potential relationships between each swimmer’s performance and their category standard, considering the changing conditions and demands associated with category transitions. For this reason, it is stated that athletic performance is assessed based on specific results within each competitive category, placing athletes within the same performance ranges while considering the adjustments in qualifying standards and access to various levels.

Regarding the differentiation by stroke specialties and preferred swimming distances, these factors were not considered in the analysis due to the very reasons mentioned: the limited sample size and the added complexity such analysis would entail. Only part of this information is included as supplementary data (this was suggested by another reviewer).

3) Discussion

The discussion begins by restating the main objectives of the study in terms that closely mirror the introduction. While briefly revisiting the aims is acceptable, this repetition adds little value unless it is immediately tied to the study’s findings. It is recommended that the authors revise the opening to more directly present and interpret their key results in light of the study’s goals.

The discussion covers a wide range of topics and successfully links the study's findings to existing research, addressing both statistically significant and non-significant outcomes. The authors appropriately acknowledge several limitations, such as the small sample size and the imbalance in gender and specialization. However, at times the section feels overly long and resembles a literature review, which can dilute the focus on the study’s own results.

It is commendable that the authors attempt to consider the combined influence of social, anthropometric, and psychological factors. Nevertheless, some of their interpretations are speculative and not clearly supported by their own data—for instance, references to the Pygmalion effect or the influence of coaching style. A more distinct separation between data-driven conclusions and those based purely on theoretical or literature-derived explanations would improve clarity.

Furthermore, the discussion would benefit from a more concise and explicit presentation of the practical implications of the findings, especially for coaches, sport program designers, and federations. Suggested directions for future research are relevant, particularly those that call for the inclusion of measures like biological maturity to better control for individual differences in development.

We have considerably shortened and focused the discussion, distinguishing between conclusions based on data and interpretations based on literature. Conversely, we have removed some assessments relating to the athlete's external perception.

4) Conclusion

The conclusion could be more concise and avoid repeating results already discussed earlier in the manuscript. It would also benefit from a clearer focus on the practical implications for coaches, training programs, and youth sport planning.

The authors’ suggestions for future research are a strength of the section. Highlighting the need for larger samples and comparative studies in other sports and age groups enhances the broader applicability of the findings. However, these more detailed research proposals might be better placed in the final part of the discussion, rather than in the conclusion itself, which should ideally focus on the key takeaways from the study.

We revised the conclusion to emphasize applied implications for coaches and removed redundancies present in earlier sections. Aspects aimed at future research directions and the practical utility of this information have also been included at the end of the discussion section.

Reviewer #2:

Dear Authors,

I want to express my gratitude for the opportunity to review this manuscript.

Congratulations on the study. Some improvements are suggested below, with line indications.

L69-70 – Please consider abbreviating “PA”.

The changes have been made by incorporating the acronyms so they can be included in the rest of the text.

L98 – “relative age” – RAE suggested.

The changes have been made by incorporating the acronyms so they can be included in the rest of the text.

L108 – Please describe the inclusion and exclusion criteria. Please describe all available information related to the subjects´ characterization. Some examples include training routines, years of experience, competitive level, and number of weekly training sessions.

The selection, inclusion, and exclusion criteria are described in the second paragraph of the Design and Procedure section.

L121-188 - Please describe all methodological details, for example, the procedures in detail, preferably with reference support. Another example is the human resources involved – academic background and experience.

The text has been slightly modified, and a related reference has been incorporated into the opening paragraph of the Instruments section. In the Design and Procedure section, additional details have been included regarding the process and the research team

L191-206 – Please consider shorter paragraphs to improve readability (8-12 lines suggested). Please make sure all statistical procedures are described.

The text has been revised and refined to enhance clarity and coherence in the discourse adding more information about procedures.

L216 – Please revise all tables´ content and format, considering the journal template and instructions for authors.

L207-282 – The reformulation of this section is suggested to become more appealing and easier for readers' interpretation. Please revise the text and tables.

The text has been revised and refined to enhance clarity and coherence in the discourse adding more information about procedures.

L286 – Only “RAE” suggested, previously in full. Please revise all manuscript.

The term has been replaced by its acronym throughout the different sections of the text.

L290-420 - Please consider improving the quality of the discussion section, namely regarding the study rationale, and with the inclusion of more references.

We have significantly adapted and focused the text, distinguishing between conclusions based on the data and interpretations based on the literature.

L420 – Please consider indicating the study limitations and suggestions for future research,

L422 - Please consider short and clear take-home messages, if possible, with practical applications/implications.

We revised the conclusion to emphasize applied implications for coaches and removed redundancies present in earlier sections. Aspects aimed at future research directions and the practical utility of this information have also been included at the end of the discussion section.

L456 - Please double-check the references format. For example: Titles in upper and lowercase; Journal in full and abbreviated.

The text has been revised in accordance with the journal's guidelines.

Reviewer #3:

General Feedback

The manuscript presents valuable original research on the relative age effect (RAE) in youth swimmers. The study research question is highly relevant, and the methodology is sound, highlighting how the RAE influences performance conditionally, which is a significant contribution. However, the paper can be strengthened by addressing some areas in different sections as suggested.

Abstract

The abstract can be made more concise while still including all essential information, especially about your novel findings (e.g., direct versus mediated effects).

A more precise revision of the abstract has been completed

Introduction

• Given your extensive literature review, are there any specific contradictory findings in previous research that your study directly aims to clarify or challenge?

Thank you for your comment. No contradictory studies on the subject of research on swimmers have been found. We wanted to test the hypothesis put forward at the end of the introduction against the sample to see if it holds true and confirms the previous findings.

• The introduction could benefit from a clearer articulation of the specific gap in the literature that this study directly aims to fill, beyond generally exploring RAE in swimming.

We have revised the introduction to more explicitly state the research gap regarding RAE in swimming within the Spanish context.

• The topic and idea are valid. However, the article reads more like a description than a true scientific analysis guided by a clearly defined theoretical framework. The theoretical framework connecting RAE to swimming performance requires enhancement, and it is important to include a comparison with similar studies on RAE.

Thank you for your comment. We have reviewed the introduction, making changes according to your suggestions.

Figures and tables

Could a graphical representation of key findings, such as the relationship between birth quartile and performance when mediated by BMI, enhance readers' understanding of your results?

New tables have been added

Methods

• Could you elaborate on your rationale for using a convenience sample and discuss any potential biases this might introduce?

• Given the acknowledged sample size limitation; (a). what steps did you take during the analysis to account for this (e.g., sensitivity analyses)? (b) Did you conduct any power analysis to determine the necessary sample size to detect expected effects?

A note was added in the text acknowledging the lack of a priori power analysis and the exploratory nature of the study due to sample access constraints.

• You have given 5 items on the "coach's subjective perception" based on ad hoc questionnaire. Could you provide a copy of the full questionnaire as supplementary material for reproducibility?

A copy of the ad hoc questionnaire template provided to the coaching staff is enclosed as a supporting document. It is important to note that the information gathered through this instrument was incorporated as supplementary data to the athletic performance outcomes, serving as an interview-based account and offering each coach’s subjective perspective on their respective group.

• The specific procedures for data collection should be further explained. For example, how were the questionnaires administered, what were the exact anthropometric measurement protocols if they deviate from standard including reference?

• Are there any other specific procedural details that you could expand upon for greater transparency and reproducibility? e.g., order of measurements, instructions given to swimmers during anthropometry etc.

The text has been slightly modified, and a related reference has been incorporated into the opening paragraph of the Instruments section. In the Design and Procedure section, additional details have been included regarding the process and the research team.

Statistical Analysis and Result

• Could you present a more detailed breakdown/post hoc analysis of the significant differences in ANCOVA. For instance, could you clarify which quartiles demonstrated significant differences, such as between Q1 and Q4, regarding the means and standard deviations of the dependent variables?

• The ANCOVA results are presented with F-values and p-values, but not the specific group means for the factors involved. Could you provide the means and standard deviations for the dependent variables across different/individual quartiles or months of birth, and training days, to better illustrate the significant differences?

• The specific statistical approach such as Tukey's HSD, FDR etc., to address the issue of multiple comparisons given the number of correlations and ANCOVA analyses performed are not explicitly stated. This represents a limitation in your statistical methodology, as multiple comparison corrections would have strengthened the reliability of your findings.

New writing has been added.

As suggested, we included post hoc comparisons following ANCOVA and provided tables with means and standard deviations by birth quartile and month. We clarified in the data analysis

---

## [Decision Letter · Decision Letter 1]

5 Sep 2025

Dear Dr. Alfonso Trinidad Morales,

We look forward to receiving your revised manuscript.

Kind regards,

Mohamed Ahmed Said, Ph.D.

Academic Editor

PLOS ONE

Journal Requirements:

Additional Editor Comments:

Dear Authors,

Thank you for your thorough changes and resubmission of your manuscript. We recognize the substantial effort you have invested in responding to the feedback from Reviewers 1, 2, and 3. The manuscript exhibits enhanced clarity, increased rigor, and improved alignment with the journal's specifications.

Upon evaluating your answers and updated content, we have determined that some issues remain inadequately handled, necessitating further rewriting prior to the paper's consideration for publication. Kindly review the comprehensive feedback provided below.

Reviewer 1 – Follow-up

Addressed by authors:

More precise delineation of the research is needed in the introduction.

Enhanced theoretical correlation between RAE and swimming performance.

Augmented literature review with revised citations.

Still needs elucidation/minor enhancements:

The incorporation of conflicting or diverse findings in the literature would enhance the impact of your study. Although no conflicts exist in swimming, contextualizing your findings within the broader realm of RAE research, encompassing other sports, can enhance the strength of your case.

Reviewer 2 – Follow-up

Addressed by authors:

Abbreviations (PA, RAE) have been standardized.

Inclusion and exclusion criteria, together with participant details, have been incorporated.

Methodological details and references have been enhanced.

The readability of lengthy paragraphs has been enhanced.

Statistical methodologies delineated with greater specificity.

Tables amended for clarity and adherence to journal format.

Discussion augmented with justification and supplementary sources.

Limitations and prospective research avenues are included.

References adjusted to conform to journal style.

Still needs elucidation/minor enhancements:

Additional information is required regarding the human resources engaged, including the academic qualifications and experience of the assessors.

Certain paragraphs in the Results section are very lengthy; more segmentation would enhance readability.

Reviewer 3 – Follow-up

Addressed by authors:

Abstract modified for brevity and incorporation of new discoveries.

Introduction amended to emphasize the Spanish swimming setting and identify the research deficit.

The theoretical framework has been elucidated.

Updated tables and post hoc ANCOVA findings with means and standard deviations included.

The application of Tukey HSD for multiple comparisons has been elucidated.

The coach perception questionnaire is included as supplemental information.

Supplementary methodological specifics included (procedures, anthropometry, data acquisition).

Discussion restructured to encompass theoretical contributions, limitations, and ramifications.

Recognition of gender disparity.

Emphasis on practical implications.

The database is included as an ancillary file.

Compliance with STROBE criteria verified.

Still needs elucidation/minor enhancements:

The justification for convenience sampling and its associated biases is recognized, albeit succinctly; kindly elaborate on this topic.

A power analysis was not performed. This must be explicitly recognized in the text, with a clear reference to the study's exploratory nature.

The coach perception questionnaire demonstrates an absence of validity and reliability evidence. This represents a methodological deficiency that must be stated candidly.

The discourse regarding similarities with other individual sports is still restricted. A concise yet clear reference to analogous RAE findings in other sports would enhance the interpretation.

Data availability: please confirm that the supplemental dataset adheres to the journal's policy, including repository deposition and accession numbers if mandated.

Cross-cutting Issue: Questionnaire Validity and Reliability

The ad hoc coach questionnaire is provided as supplementary material, but the authors do not report any validity or reliability testing.

This must be explicitly acknowledged as a limitation. At minimum:

State that the instrument has not undergone formal validation.

Clarify that data from it is exploratory and supplementary only.

If possible, report internal consistency (Cronbach’s α) or content validation by experts.

We urge you to amend the manuscript accordingly and resubmit it. Kindly furnish a comprehensive, point-by-point response letter delineating how you have resolved each of the unresolved issues.

Respectfully,

Dr. Mohamed Ahmed Said

Reviewers' comments:

Reviewer's Responses to Questions

**Comments to the Author**

Reviewer #3: All comments have been addressed

2. Is the manuscript technically sound, and do the data support the conclusions?

Reviewer #3: Yes

3. Has the statistical analysis been performed appropriately and rigorously?

Reviewer #3: Yes

4. Have the authors made all data underlying the findings in their manuscript fully available?

Reviewer #3: Yes

5. Is the manuscript presented in an intelligible fashion and written in standard English?

Reviewer #3: Yes

Reviewer #3: I am pleased by the keen response by authors to the issues raised in the first round of review. I am satisfied with the responses given to my comments and their integration within the text of the manuscript.

**Do you want your identity to be public for this peer review?** For information about this choice, including consent withdrawal, please see our Privacy Policy

Reviewer #3: No

---

## [Author Response · Author response to Decision Letter 2]

30 Sep 2025

Response Letter to the Reviewers – Manuscript “Relative age of youth swimmers and their sporting performance at the end of the season”

Dear Dr. Mohamed Ahmed Said,

Academic Editor of PLOS ONE,

Dear Reviewers,

We deeply appreciate your valuable comments and the time you have dedicated to reviewing our manuscript. We are grateful for your acknowledgment in your recent communication that we have corrected, modified, and incorporated nearly all of the suggestions and improvement recommendations provided by the reviewers in the previous round.

Below, we present a new point-by-point response to each of the comments received, indicating the new modifications made to the previously revised version of the manuscript and justifying those that could not be implemented. Modifications and changes in citation numbering within the text have been highlighted in yellow. New references have been added to the corresponding section and are also highlighted in yellow.

We hope this clarifies the points that remained pending in the reviewers’ judgment.

Reviewer 1

Comment: It is suggested to incorporate contradictory or diverse findings from the literature to enrich the impact of the study.

Response: We have expanded the literature review previously modified, including studies on the Relative Age Effect (RAE) in other individual sports such as athletics, tennis, taekwondo, and gymnastics, where divergent results have been observed (lines 371–377; 385–389; 408–410; 425–428; 430–436). This contextualization has been added to the introduction and discussion sections to strengthen the interpretation of our findings in swimming. The new references have been added to the final section and are highlighted in yellow.

Reviewer 2

Comment: Additional information is required regarding the human resources involved, including academic background and experience of the evaluators.

Response: A detailed description of the academic profile and professional experience of the evaluators was already included in the methodology section of the previous version. In this version, we have expanded it with additional data (lines 219–226).

Comment: Some paragraphs in the results section are lengthy.

Response: We have further segmented the section into shorter and more distinct paragraphs to improve readability and facilitate understanding, as recommended. Additionally, in previous versions of this manuscript, a reviewer had suggested expanding certain explanations, which we did in accordance with that recommendation.

Reviewer 3

Comment: Justification for convenience sampling.

Response: We have expanded the justification for convenience sampling in the methodology section, explaining its inherent limitations and potential impact on the generalizability of the results (lines 127–137).

Comment: No power analysis was conducted.

Response: We have added an explicit statement in the limitations section indicating that no power analysis was conducted due to the exploratory nature of the study and the sample size (lines 476–481). Reliability evidence is provided in the instruments section.

Comment: The coach perception questionnaire lacks evidence of validity and reliability.

Response: We have acknowledged this limitation in the methodology/instruments and discussion sections, indicating that the instrument has not been formally validated and that the data obtained should be considered exploratory. Additionally, we have added a note on the need for future studies to validate this questionnaire. Reliability evidence has been added (Cronbach’s Alpha and McDonald’s ω, lines 180–185). These data will be included as a supplementary document for reviewer consultation in the journal’s annexes.

Comment: Limited discussion on similarities with other individual sports.

Response: We have expanded the discussion to include references to studies on RAE in sports such as judo, taekwondo, gymnastics, and athletics, to enrich the interpretation of the results—both in terms of confirming the RAE and studies indicating that some results are not definitive (lines 371–377; 385–389; 408–410; 425–428; 430–436). The new references have been added to the final section and are highlighted in yellow.

Comment: Confirm that the supplementary dataset complies with the journal’s policy.

Response: We confirm that the dataset related to this work has been provided to the journal for reviewer and editor consultation, ensuring access and availability should it be required.

Cross-cutting Issue: Validity and Reliability of the Questionnaire

Comment: The questionnaire has not been validated.

Response: We have added an explicit statement in the methodology/instruments and discussion sections acknowledging that the questionnaire has not been validated in previous studies. However, it has undergone reliability testing (Cronbach’s Alpha and McDonald’s ω). These data have been incorporated into the text (lines 180–185; 476–481; 512–517). The data will be included as a supplementary document for reviewer consultation if desired. We indicate that the data obtained are exploratory and suggest conducting future studies to validate the instrument. The limited sample size restricts the ability to detect stronger internal consistency, but this limitation has been noted as a methodological constraint.

We reiterate our gratitude for your observations, which have significantly contributed to improving the quality and rigor of our work.

We remain attentive to any further suggestions you may consider appropriate.

Sincerely,

Dr. Daniel Mendoza Castejón, on behalf of the research team.

---

## [Editor Report · Decision Letter 2]

6 Oct 2025

Relative age of youth swimmers and their sporting performance at the end of the season

PONE-D-25-21539R2

Dear Dr. Alfonso Trinidad Morales,

We’re pleased to inform you that your manuscript has been judged scientifically suitable for publication and will be formally accepted for publication once it meets all outstanding technical requirements.

Kind regards,

Mohamed Ahmed Said, Ph.D.

Academic Editor

PLOS ONE
---

## [Editor Report · Acceptance letter]

PONE-D-25-21539R2

PLOS ONE

Dear Dr. Trinidad Morales,

I'm pleased to inform you that your manuscript has been deemed suitable for publication in PLOS ONE. Congratulations! Your manuscript is now being handed over to our production team.

Kind regards,

on behalf of

Dr. Mohamed Ahmed Said

Academic Editor

PLOS ONE